# *Pneumocystis jirovecii* Pneumonia after Heart Transplantation: Two Case Reports and a Review of the Literature

**DOI:** 10.3390/pathogens12101265

**Published:** 2023-10-21

**Authors:** Carlo Burzio, Eleonora Balzani, Silvia Corcione, Giorgia Montrucchio, Anna Chiara Trompeo, Luca Brazzi

**Affiliations:** 1Department of Anesthesia, Intensive Care and Emergency, Città della Salute e della Scienza di Torino Hospital, 10126 Torino, Italy; c.burzio@gmail.com (C.B.); giorgiagiuseppina.montrucchio@unito.it (G.M.); atrompeo@cittadellasalute.to.it (A.C.T.); luca.brazzi@unito.it (L.B.); 2Department of Surgical Science, University of Turin, 10124 Torino, Italy; 3Department of Medical Sciences, Infectious Diseases, University of Turin, 10124 Turin, Italy; silvia.corcione@unito.it; 4School of Medicine, Tufts University, Boston, MA 02111, USA

**Keywords:** solid organ transplant, heart transplant, lung transplant, opportuinistic infections, fungal infections, *Pneumocystis jirovecii*

## Abstract

Post-transplant *Pneumocystis jirovecii* pneumonia (PcP) is an uncommon but increasingly reported disease among solid organ transplantation (SOT) recipients, associated with significant morbidity and mortality. Although the introduction of PcP prophylaxis has reduced its overall incidence, its prevalence continues to be high, especially during the second year after transplant, the period following prophylaxis discontinuation. We recently described two cases of PcP occurring more than one year after heart transplantation (HT) in patients who were no longer receiving PcP prophylaxis according to the local protocol. In both cases, the disease was diagnosed following the diagnosis of a viral illness, resulting in a significantly increased risk for PcP. While current heart transplantation guidelines recommend *Pneumocystis jirovecii* prophylaxis for up to 6–12 months after transplantation, after that period they only suggest an extended prophylaxis regimen in high-risk patients. Recent studies have identified several new risk factors that may be linked to an increased risk of PcP infection, including medication regimens and patient characteristics. Similarly, the indication for PcP prophylaxis in non-HIV patients has been expanded in relation to the introduction of new medications and therapeutic regimens for immune-mediated diseases. In our experience, the first patient was successfully treated with non-invasive ventilation, while the second required tracheal intubation, invasive ventilation, and extracorporeal CO2 removal due to severe respiratory failure. The aim of this double case report is to review the current timing of PcP prophylaxis after HT, the specific potential risk factors for PcP after HT, and the determinants of a prompt diagnosis and therapeutic approach in critically ill patients. We will also present a possible proposal for future investigations on indications for long-term prophylaxis.

## 1. Introduction

Infections are the primary cause of death within 1 year of heart transplantation, accounting for 32% of deaths during this period [1]. Although they occur at a lower incidence in the subsequent years, infections continue to be a significant cause of death, primarily represented by nosocomial infections, reactivations, and opportunistic infections during the early post-transplantation period [1].

The increasing indications for organ transplantation, including higher age limits and sicker patients, accentuate the incidence of post-transplant infectious complications [2]. Among all opportunistic infections, fungal infections after organ transplantation continue to be a leading cause of morbidity and mortality [3].

*Pneumocystis jirovecii* pneumonia (PcP) is a rare opportunistic infection initially identified in HIV patients [1]. Subsequently, the role of *Pneumocystis jirovecii* as a pathogen in immunocompromised non-HIV patients has been established [4]. This infection is referred to as PcP (Pneumocystis pneumonia), but the abbreviation PJP (referring to the renaming of the pathogen as *Pneumocystis jirovecii*) is also commonly used, although formally incorrect [5]. PcP is a well-known complication in clinical cases characterized by a decreased immunological response, and adequate prophylaxis has been proposed for various populations of immunocompromised hosts, including HIV patients and solid organ transplantation (SOT) recipients [6,7]. After an initial decline in the incidence of PcP infection due to the widespread implementation of prophylaxis regimens, there has been a re-emergence of this opportunistic infection in the SOT recipient population [8,9,10]. Furthermore, the recent introduction of new immunomodulation drugs has been associated with an increase in PcP cases in non-HIV patients, leading to the advocacy of new prophylaxis regimens [11,12,13,14].

We recently treated two cases of *Pneumocystis jirovecii* pneumonia (PcP) occurring in heart transplantation (HT) recipients in the Cardiosurgical Intensive Care Unit (ICU) of the University Hospital ‘Città della Salute e della Scienza’ in Turin, Italy. PcP caused severe respiratory failure in both patients, necessitating ICU admission. Based on this clinical experience, we conducted a review of the available literature to highlight the epidemiology of late PcP in HT recipients and describe the relative prophylaxis recommendations. Here, we report the determinants of a timely diagnosis and the therapeutic approach in HT recipients, as well as some new proposals for appropriate alternative timing regimens.

## 2. Clinical Cases

### 2.1. First Case

The first case involved a 62-year-old man with a history of post-ischemic cardiomyopathy and progressive heart failure with reduced ejection fraction, who underwent an orthotopic heart transplantation. The patient presented to the post-transplant clinic 14 months after the transplant with progressive weakness and worsening shortness of breath. Laboratory testing revealed moderate lymphocytopenia. A high-resolution computed tomography (HRCT) scan was performed, revealing diffuse “ground-glass” thickening in both lungs (Figure 1B).

Admission laboratory results and imaging findings are shown in Table 1.

A diagnostic bronchoscopy and bronchoalveolar lavage (BAL) were performed, resulting in a positive test for PJ-DNA; for rapid *Pneumocystis jirovecii* detection, Eazyplex *Pneumocystis jirovecii* kit on Amplex Genie II platform (AmplexDiagnostics, GmbH, Gars am Inn, Germany) were used directly from bronchoalveolar lavage specimen. Despite the timely initiation of intravenous trimethoprim-sulfamethoxazole (TMP-SMX) therapy (4 days after hospitalization), the patient developed moderate acute respiratory failure, requiring ICU admission. Simultaneously, the immunosuppression regimen was modified by discontinuing mycophenolate (stopping for the first week, then 250 mg BID) and increasing steroid therapy (switch to IV Methyl-Prednisone, 125 mg TID). Further investigation revealed mild cytomegalovirus (CMV) viremia (CMV-DNA 4400 IU/mL), which led to the initiation of pre-emptive therapy using valganciclovir. The patient responded rapidly to treatment, and his condition improved within 48 h of ICU admission. His treatment course included two weeks of intravenous TMP-SMX at a dose of 800/160 mg twice daily, followed by one week of step-down therapy with oral TMP-SMX at the same dosage. Secondary prophylaxis with oral TMP-SMX at a dose of 400/80 mg once a day was then initiated. After a total of 24 days of hospitalization, the patient was successfully discharged home.

### 2.2. Second Case

The second case involved a 60-year-old man who had undergone a successful heart transplantation 8 years earlier and was hospitalized due to respiratory failure. The patient presented to the Emergency Department with complaints of high-grade fever (38 °C), shortness of breath, cough, and malaise, which had been recurring over the past two weeks. Arterial blood gas analysis showed moderate respiratory failure. A chest CT scan revealed diffuse bilateral ground-glass opacities, most prominent in both apical lobes (Figure 1B). Despite receiving broad-spectrum antibiotic therapy for suspected bacterial infection, the patient failed to improve. Quantitative serum Serum Beta-D-Glucan (BDG) was performed (ß-Glucan Test on Toxinometer MT-6500, FUJIFILM Wako Pure Chemical Corporation, Japan) and resulted positive (207.7 pg/dL, reference value < 7pg/dL). Testing on a BAL sample (Eazyplex PJ kit, Amplex Diagnostic, Gars am Inn, Germany). Serum Beta-D-Glucan was positive. PCR testing on a BAL sample resulted in a positive test for PJ-DNA and for *Herpes simplex* Virus (HSV) DNA (210,500 copies/mL). Suspecting PcP secondary to HSV reactivation, a regimen was initiated including trimethoprim/sulfamethoxazole (intravenous, 800/160 mg three times a day), steroids (dexamethasone, intravenous, 4 mg three times a day), and acyclovir (intravenous, 500 mg three times a day), three days after hospital admission. Additionally, the immunosuppression therapy was reduced by lowering the dosage of ciclosporin and discontinuing mycophenolate. Despite the microbiological diagnosis, the patient’s respiratory condition evolved into severe acute respiratory distress syndrome (ARDS). Initially treated with non-invasive ventilation, his respiratory failure did not improve despite invasive ventilation, pronation cycles, and extracorporeal CO2 removal (ECCO2R). After 7 days, a re-evaluation of treatment was deemed necessary. Microbiological examinations were repeated, documenting an increase in BDG and no evidence of other opportunistic infections. Meanwhile, broad-spectrum antibiotics were empirically added, and antifungal therapy was escalated using caspofungin (intravenous, loading dose 70 mg, followed by 50 mg/day) in combination with the previous dosage of TMP-SMX. However, the patient’s condition continued to deteriorate until he passed away 18 days after ICU admission.

## 3. Epidemiology of PcP

### 3.1. Incidence of PcP

The incidence of PcP in SOT recipients has varied over time due to the introduction of early prophylaxis in the post-operative period. Historically, the highest risk period for PcP was within the first six months after transplantation, leading to guidelines recommending early prophylaxis during this period [4]. The implementation of routine prophylaxis during the first 6 to 12 months after transplantation has significantly reduced the onset of PcP within the first-year post-transplantation. However, during the second year after transplantation, when prophylaxis is no longer commonly recommended, the risk of PcP infections increases. Notably, patients who received TMP-SMX prophylaxis during the first year have a higher risk of PcP infection in the subsequent period if prophylaxis is discontinued [4,7].

Numerous studies have indicated that post-transplant PcP is a re-emerging infection among SOT recipients, especially after discontinuation of prophylaxis, and it is associated with significant morbidity and mortality [6,8,9]. Studies involving mixed populations of organ transplant recipients have reported on the incidence of PcP, but HT recipients are often under represented [10,11,12].

In a nationwide prospective cohort study with an 8-year follow up, Neofytos et al. studied the incidence rate of both early and late PcP in a population of SOT recipients, including 204 HT recipients (7.2%) [6]. Notably, HT recipients showed a higher incidence (0.02/1000 person-days) of PcP than recipients of other SOTs (0.02/1000 person-days), and 60% of PcP cases in HT recipients occurred more than one year after transplantation [6].

### 3.2. Risk Factors for PcP in HT Recipients

Data from retrospective studies show that a high number of PcP cases occur in non-HIV patients who have no indication for prophylaxis but have known risk factors for PcP infection, such as malignancy, autoimmune disease, and prior SOT [13]. This raises the question of whether expanding the indications for prophylaxis could prevent a significant proportion of PcP cases [13]. Several studies have investigated risk factors linked to PcP development, although most of the evidence comes from studies involving heterogeneous populations of SOT recipients.

Iriart et al. examined 33 cases of PcP in SOT recipients that occurred one year after transplantation and after the termination of prophylaxis. The results of the multivariate analysis suggested that age > 65 years, detectable cytomegalovirus (CMV) viremia, and a total lymphocyte count < 750/mm^3^ persisting for more than one month were significant risk factors for the development of PcP and were proposed as indications for extended prophylaxis [4].

The study by Neofytos et al., which enrolled a national cohort of SOT recipients, including 204 HT recipients, identified age ≥ 65 years and CMV infection during the first 6 months post transplant as significant predictors of late-onset PcP [6].

A subsequent systematic review and meta-analysis by Permpalung et al. included 30 studies involving over 400,000 SOT recipients [12]. However, only two of these studies involved HT recipients. Factors associated with PcP development included acute rejection, CMV-related disease, absolute lymphocyte count < 500 cells/mm^3^, BK polyomavirus-related diseases, HLA mismatch ≥ 3, rituximab use, and polyclonal antibody use for rejection. The authors suggested considering expanded PcP prophylaxis, in addition to the criteria already in use, in SOT recipients with lymphocytopenia, BK polyomavirus-related infections, and rituximab exposure [12].

Interestingly, beyond lymphocytopenia, which is one of the most studied and recognized risk factors for PcP, the role of CD4 lymphocytes in the pathogenesis of non-HIV related PcP has recently been acknowledged [15,16]. One retrospective study had demonstrated that a low level of CD4 lymphocytes is correlated with the development of PcP. Furthermore, in two retrospective studies involving non-HIV PcP patients, low CD4 lymphocyte levels appear to be associated with a worse prognosis [15,16].

In Figure 2, we have incorporated all the risk factors and grouped them by subgroups in a forest plot.

### 3.3. Viral Infection as Risk Factor for PcP

Recently, the role of viral infections in the pathogenesis of PcP has been increasingly discussed, with particular attention given to the role of viruses as co-infections or risk factors. Among these viruses, CMV has been extensively studied and identified as a significant risk factor and comorbidity associated with increased mortality in PcP among non-HIV patients [14,17]. Furthermore, numerous studies and meta-analyses conducted in the SOT recipient population have consistently demonstrated CMV as one of the main risk factors for PcP development [6,11,12].

Although the role of CMV in PcP is well established, the clinical significance of other viral infections is yet to be defined. In particular, the relationship between SARS-CoV-2 (severe acute respiratory syndrome COVID-19) and respiratory superinfections, such as COVID-19-associated pulmonary aspergillosis (CAPA), has recently gained considerable interest [18]. Case reports have highlighted occurrences of PcP in patients, both immunocompromised [19,20] (including SOT recipients) [21], and previously healthy patients without risk factors other than SARS-CoV-2 infection [22,23].

Furthermore, one case series described five patients hospitalized for COVID-19 pneumonia who developed PcP during their hospital stay, and only one of them was an immunocompromised host. These reports have suggested a common association between significant lymphocytopenia (<200/mm^3^) following SARS-CoV-2 infection and treatment with high-dose steroids for at least two weeks [24,25]. Although the true relationship between SARS-CoV-2 and PcP requires further investigation due to limited data, comorbidities secondary to COVID-19, particularly lymphocytopenia and the use of immunosuppressive therapies, appear to be significant predisposing factors for PcP development [25].

Similar considerations have been made in the past regarding the H1N1 influenza epidemic. Several case reports have described patients with H1N1 influenza who subsequently developed PcP co-infection [26,27,28,29]. In these cases, the presence of multiple immunosuppressive factors, including secondary lymphocytopenia, was suspected to be an important predisposing factor.

Although the evidence is limited to case reports and case series, *Herpesviridae* co-infections, such as *Herpes simplex* virus (HSV) and Varicella-zoster virus (VZV), have also been implicated in the pathogenesis of PcP [30,31]. In these patients, the role of steroid therapy appears to play an important role as a predisposing factor. However, in a retrospective study involving 70 non-HIV patients affected by PcP, it has been demonstrated that HSV co-infection worsens the prognosis with an odds ratio similar to that of CMV [32].

Although studies in this area are still limited, the scientific community has nevertheless hypothesized the existence of a co-pathogenesis involving respiratory viral and fungal co-infections [33]. More in detail, it has been postulated that this role goes beyond lymphocyte count or the administration of immunosuppressive drugs and includes complex mechanisms of interaction, including epithelial barrier loss and changes in both humoral and cell-mediated immune responses [33]. A recent study on the lung microbiome in SOT recipients suggests that viral colonization or co-infection is an important factor influencing prognosis, regardless of the primary cause of infection. In this context, CMV seems to play a primary role [34].

### 3.4. Immunosuppressive Therapies as Risk Factor for PcP

Numerous cases of PcP among immunocompromised patients, who do not fall into the traditionally recommended groups for PcP prophylaxis such as HIV patients and SOT recipients, have been reported. This has led to the proposal of new indications for prophylaxis in various categories, including hematologic and onco-hematologic patients [35], patients with autoimmune and inflammatory diseases [36], patients with neurological diseases [37], and patients with inflammatory bowel diseases [38].

Particularly strong evidence exists supporting the risk of PcP in non-HIV patients regarding the use of immunosuppressive medications; in particular, the use of high-dose and long-term corticosteroids has been associated with the risk of PcP in non-HIV patients. Recommendations for non-oncologic hematologic patients suggest considering PcP prophylaxis for patients requiring high-dose corticosteroids (>30 mg/day prednisone equivalent [PEQ]) for at least 4 weeks [35,36,37,38,39], medium-dose corticosteroids (15–30 mg/day PEQ) for at least 8 weeks [35,36,37,38,39,40], or low-dose corticosteroids (>10 mg/day PEQ) in the presence of additional risk factors (age > 65 years, parenchymal lung disease, additional immunosuppressive drugs) [35,40]. Interestingly, the majority of SOT recipients fall into at least one of these categories.

The impact of several different immunosuppressive medications has been studied, including regimens widely used in SOT recipients such as Mycophenolate Mofetil (MMF) and Mycophenolate Acid (MPA), which inhibit the proliferation of B and T lymphocytes. In two different case-control studies, MPA was associated with an increased incidence of PcP [38,39], and both studies included liver or renal transplantation recipients. A recent retrospective study involving patients suffering from connective tissue disease showed that a large proportion of these patients who later developed PcP had been treated with MMF [41]. Furthermore, in this study, MMF use was found to be associated with increased mortality. Calcineurin inhibitors, such as ciclosporin and tacrolimus, and mTOR inhibitors, mainly sirolimus and everolimus, are also widely used in immunotherapy among SOT recipients. Studies attempting to demonstrate a clear association of Calcineurin inhibitors with PcP have produced conflicting results [4]. Nevertheless, immune-therapy regimens based on the use of more than one medication have been found to be at higher risk for PcP in SOT recipients. Among these, regimens including mTOR inhibitors seem to be particularly at risk [42].

The interaction between other immunosuppressive drugs and PcP has been less extensively studied, especially considering the large number of new drugs introduced in recent years. Recent evidence has linked the use of rituximab to the development of PcP [43,44,45]. A case-control study involving 76 cases and 159 controls, all belonging to the non-HIV patient population, demonstrated that rituximab treatment is an important independent risk factor for the development of PcP [43]. Subsequently, a follow-up study showed that primary PcP prophylaxis in patients receiving rituximab treatment significantly reduced the incidence of infections with a good safety profile [44]. These data are particularly relevant as SOT recipients may undergo rituximab treatment, especially in cases of antibodies-mediated rejection and post-transplant lymphoproliferative disorder (PTLD) [45]. PTLD has also been associated with the development of PcP, but it is unclear whether this risk is solely attributable to pharmacological treatment (including immunosuppressive drugs and often rituximab) or if there are additional elements of interplay between immunosuppressive therapy and immune response [45].

### 3.5. PcP Prophylaxis: State of the Art

The International Society for Heart and Lung Transplantation (ISHLT) guidelines recommend universal PcP prophylaxis for the first 6–12 months following HT [46]. The duration of prophylaxis may vary among different medical centers, with some centers recommending as little as 6 months and others suggesting 12 months or even lifelong prophylaxis [47].

In addition to primary prophylaxis, the guidelines also suggest considering extended PcP prophylaxis for individuals with chronic CMV infection, prior PcP episodes, or those requiring augmented immunosuppression [46].

Nevertheless, there is considerable variability in the application of these recommendations. A recent French survey has demonstrated that, even within the same national transplant network, there is a significant variability in PcP prophylaxis, particularly concerning the dosage used and the duration of therapy [48].

Additional recommendations for extended prophylaxis have been proposed based on known risk factors for PcP. These recommendations primarily focus on evaluating recipients’ white blood cell counts and considering different alternative immunosuppression therapies, including high-dose regimens and combination regimens. Lifelong PcP prophylaxis is also being proposed [49,50].

The universal, expanded, and proposed criteria are summarized in Table 2.

## 4. Clinical Features of PcP

### 4.1. Clinical Manifestation and Diagnosis of PcP

Fever, non-productive cough, and dyspnea are typical signs of PcP in non-HIV patients, including immunocompromised individuals such as HT recipients. These symptoms usually develop gradually over the course of 1 or 2 weeks and are often associated with hypoxemia. Respiratory failure is a common event but can manifest at various degrees during the clinical course, making it highly nonspecific. Classically, hypoxia is disproportionate to radiographic imaging, and the clinical course is typically more severe in non-HIV patients [47,51,52].

### 4.2. Imaging of PcP

PcP is a diffuse interstitial pneumonia, and no specific radiographic chest X-ray pattern is pathognomonic for *Pneumocystis* infection. In lung transplantation (LT) recipients receiving pentamidine prophylaxis, a classical presentation is bilateral upper lobe infiltration, but the most commonly described radiographic pattern in PcP is characterized by bilateral diffuse ground-glass opacities with interstitial infiltrate [47].

However, the radiographic pattern of PcP can be inconsistent [53]. A recent study involving 360 patients with clinical suspicion of PcP found that no X-ray finding, other than increased interstitial markings and radiologist impression, is significantly associated with a PcP diagnosis [54]. Moreover, 10–15% of PcP patients (in non-HIV patients) have a normal chest radiography, and up to 30% present with non-specific radiographic features [53]. Overall, plain chest radiography is a poor diagnostic tool.

High-resolution CT (HRCT) scan appears to have a better diagnostic yield. Typical findings include ground-glass opacities (GGOs), which are usually symmetric, predominant in the perihilar regions and apices, with peripheral sparing (seen in approximately 20% of cases), or a diffuse mosaic pattern (up to 60% of cases) [55]. Features of advanced disease include intralobular lines superimposed onto ground-glass opacities (referred to as a “crazy paving” aspect) and lung consolidation, with these features being more common in non-HIV patients [55,56]. Nodules and/or septal thickening are other findings consistent with a PcP diagnosis [54].

Although the radiographic features of PcP have been extensively investigated, they encompass a wide range of common and less frequent findings, along with a broad spectrum of differential diagnoses. Therefore, imaging cannot provide a definitive diagnosis but can offer essential clues to guide the diagnostic approach [54,55,56].

The two cases we observed greatly differ regarding imaging characteristics, particularly the HRCT features (Figure 1). The first case had typical HRCT findings, including diffuse symmetrical ground-glass opacities with apical predominance and peripheral sparing. In contrast, the second case exhibited a less distinct radiographic pattern, with diffuse ground-glass opacities resembling a mosaic pattern at some points but lacking apical predominance and peripheral sparing, with only mild perihilar enhancement.

### 4.3. Laboratory Testing

Laboratory testing can be helpful in diagnosing PcP, with lactic dehydrogenase (LDH) and Beta-D-Glucan (BDG) being recommended, although their utility may vary [47]. Serum LDH levels are often elevated (>300 IU/mL) in PcP, but this elevation is not specific and can occur in many other underlying diseases, including other causes of acute lung injury. Recent studies have shown that LDH levels are generally not elevated in non-HIV patients with PcP. In one retrospective study, LDH elevation was observed in only 40% of non-HIV PcP patients compared to 73% of HIV-positive patients [57]. Furthermore, in another retrospective comparison, LDH elevation had a sensitivity of 63% and a specificity of 43% for diagnosing PcP in non-HIV patients, with significantly different values in the HIV cohort [58]. Therefore, elevated LDH levels may raise suspicion of PcP but have limited utility for a firm etiologic diagnosis and as a diagnostic criterion for PcP in non-HIV populations [47,57,58].

Serum Beta-D-Glucan (BDG) assays are commonly available for evaluating possible fungal infections [59,60]. Beta-D-Glucan is a cell wall component found in many fungal species, including the cyst form of *Pneumocystis* [61]. This has led to the suggestion of using BDG assays for diagnosing PcP. However, since BDG is a common component among various fungal species, there is a theoretical risk of encountering false positives [62]. It is important to note that the most common fungal species that exhibit BDG, such as the *Candida* genus, typically do not cause respiratory pathology, even in immunocompromised patients [62,63]. On the other hand, the most common respiratory fungal infections are caused by hyphae-forming fungi, such as *Aspergillus*, which might be associated with a negative BDG test [62,63]. Additionally, several pathogens that express BDG can cause respiratory infections [64], including *Histoplasma* [65,66], *Fusarium* [67,68], and *Trichoderma* [69,70], but these are rare pathogens, even in immunocompromised hosts [64].

Overall, a meta-analysis involving both HIV and non-HIV patients found that the BDG test had a sensitivity of 94.8% and a specificity of 86.3% for PcP, with a high negative predictive value [71]. However, another meta-analysis reported that the sensitivity of the BDG test is lower (86%) in non-HIV patients, with comparable specificity. In this study, a negative BDG test was associated with a low post-test probability of PcP (≤5%) only when the pre-test probability was low to intermediate (≤20% in non-HIV and ≤50% in HIV) [72]. Furthermore, the usefulness of serologic testing had been questioned in patients undergoing immune-suppressive therapy and in the presence of false-positive associated factors, such as the use of cellulose-derived devices and products, the application of glucan-containing gauzes, the presence of Gram-negative endotoxemia, and the use of some antimicrobials.

In a retrospective study, the two markers previously described (LDH and BDG) were compared with two other serological markers for the diagnosis of PcP, namely Krebs von den Lungen 6 antigen (KL-6) and S-Adenosyl-Methionine (SAM) [73]. KL-6 is a host molecule, strongly expressed on type II alveolar pneumocytes and bronchiolar epithelial cells; KL-6 was linked to interstitial lung disease rather than being a specific marker of PcP [73]. SAM is an endogenous metabolic intermediate that is required in the metabolism of at least some strains of Pneumocystis [73]. In this study, BDG was found to be the most reliable serologic biomarker for PcP diagnosis [73]. Interestingly, in this study, the combination of BDG and KL-6 was the most accurate serologic approach (94.3% sensitivity and 89.6% specificity) and was therefore proposed as a minimally invasive diagnostic approach [73].

Although the BDG test cannot be used for the definitive diagnosis of PcP, nor as a safe exclusion tool for patients with acute respiratory failure with significant immunocompromised status [62,72], this biomarker maintains a high negative predictive value. Indeed, in patients whose clinical presentation raises a high index of suspicion, the BDG test can serve as a rapid, useful, and minimally invasive diagnostic tool, helping to raise suspicion of PcP infection and identify patients in need of further investigation. Finally, it may be a useful tool for presumptive diagnosis in patients who cannot obtain invasive specimens safely [51].

### 4.4. Microbiological Diagnosis and Diagnostic Approach of PcP in HT Recipients

Traditionally, microbiological diagnosis of PcP is considered challenging due to the inability to culture the microorganism in standard culture media [74]. *Pneumocystis jirovecii* can grow in vitro on selected terrains, but these systems are complex, expensive, and not useful for routine use [75,76,77]. Only very recently, a stable PJ culture was developed, using an axemic medium system; while further optimization of the culture conditions is needed, this approach is promising for obtaining PJ cultures for clinical purposes [78].

Classically, PcP diagnosis was confirmed via direct visualization of the pathogen via staining. [70] While these tests are easy and cheap to perform, they lack sensitivity due to dependence on the quality of the sample and observer interpretation, particularly when fungal burden is low, as in non-HIV patients [74]. However, they maintain a high-grade recommendation in most guidelines, due to the robust supporting literature (albeit mostly regarding the HIV population) [47].

Immunofluorescence, introduced later and using monoclonal antibodies to *Pneumocystis jirovecii*, also known as direct immunofluorescent antibodies (DFA), appears to have greater sensitivity [51]. Other methods, using polymerase chain reaction (PCR), appear to have higher sensitivity [74]. The introduction of real-time PCR or quantitative PCR (qPCR) allowed rapid quantitative diagnostic results [75]. Both methods can be performed on sputum and deep samples such as bronchoalveolar lavage (BAL) or transbronchial biopsy and are becoming increasingly important in the diagnosis of PcP [74].

Several new methods to diagnose PcP have been proposed, including antibodies assays, new targets for PCR, and loop mediated isothermal amplification (LAMP) [74]. LAMP provides an alternative to PCR methods, as it can amplify a target gene with only a heating device and isothermal conditions [79]. In both HIV and non-HIV populations, LAMP had demonstrated high sensitivity (84%–99%) and specificity (96%–99%), comparable to PCR methods [74,79,80,81]. However, in one study, a LAMP quantification method, known as time to positivity (TTP), showed a worse correlation to fungal load than cycle threshold (Ct) of qPCR, which remains the gold standard [81].

The current reference specimen for PcP diagnosis is BAL, which is considered the highest quality respiratory specimen and yields better diagnostic accuracy in non-HIV patients [33]. However, bronchoscopy carries a greater risk to the patient and may not be feasible in patients with respiratory failure. Non-invasive specimens, such as sputum testing and oral washing, are easier and often safer specimens to obtain but are considered less accurate than invasive sampling [74,82] and should be carefully considered in non-HIV patients [82]. In patients for whom pathogen identification through respiratory samples does not allow quantitative diagnosis, the combined use of qPCR and serum BDG has been proposed [83,84,85].

Because first-line diagnostic tools overall have a low predictive value for the diagnosis of PcP, it is important for clinicians to have a high level of suspicion in at-risk patients. These include SOT recipients who have completed universal prophylaxis and exhibit increased forms of immunosuppression, such as leukopenia, lymphopenia, high-dose, or multiple immunosuppressive drug regimens, multiple rejection episodes, or significant CMV viremia [47]. Clinicians should not underestimate additional elements of immunosuppression which, although not included among the classical risk factors for PcP, can influence the immune status of these patients. This group includes advanced age, viral diseases other than CMV, the epidemiological situation in which the patient finds himself (given PcP’s tendency to occur in clusters), and the presence of multiple risk factors [6,51,86].

As with other fungal diseases, diagnostic criteria have been proposed to improve the diagnostic approach to PcP. The established criteria for the definition of PcP, proposed by EORTC/MSGERC [87] and a proposed approach to PcP diagnosis, based on previous studies, are shown in Figure 3.

### 4.5. Therapeutic Approach

Most data on the pharmacological therapy approach for PcP come from studies on HIV patients, while additional evidence is derived from retrospective studies on non-HIV cohorts. From these data, therapeutic guidelines for non-HIV patients have been extrapolated [47,88].

In accordance with the strongest evidence and as endorsed by current guidelines, the first-line agent and treatment of choice is high-dose trimethoprim-sulfamethoxazole (TMP-SMX) at high doses (15–20 mg/kg/day of TMP component, divided into three or four doses) [47,88]. The use of therapeutic drug monitoring (TDM) of TMP-SMX, although not universally supported in the literature [89] and not always available, may be indicated in some patients [90,91]. In these cases, TDM must ensure a peak plasma target concentration of the sulfamethoxazole component within the range of 100–200 mg/L. Interestingly, retrospective data suggest that a low-dose (from 7.5 to 15 mg/kg/day) TMP-SMX regimen could be equally effective, especially in non-HIV patients, but burdened with fewer adverse effects [92,93,94]. Two subsequent meta-analyses, based only on retrospective data, seem to confirm this result [95,96]. A randomized controlled trial assessing low-dose TMP-SMX for PcP is currently enrolling [97]. In the meantime, an approach involving step-down therapy to low-dose TMP-SMX continues to be recommended [98].

Alternative first-line regimens for patients with contraindications to TMP-SMX include intravenous (IV) pentamidine (4 mg/kg/day), the combination of primaquine (15–30 mg/day) + clindamycin (600–900 mg q6-8h), and atovaquone (750–1500 mg bid). Regarding atovaquone and primaquine + clindamycin, their use has only been studied in mild-to-moderate PcP and only through retrospective studies in which the non-HIV population was poorly represented [99,100]. Transitioning to second-line therapies should be carefully considered, as their efficacy is less studied compared to first-line agents [88,101].

The recommended duration of treatment is 3 weeks, although a duration of 2 weeks may be considered for mild infections [47,88]. These recommendations derive from the HIV population, as the impact of treatment duration has not been evaluated in non-HIV patients [52,102]. Careful consideration should be given to patients who require secondary prophylaxis [47].

In both presented cases, as they were critically ill and immunosuppressed patients, the prudent choice was to use IV TMP-SMX at high doses as first-line therapy. Being critically ill and immunosuppressed patients, this approach seems cautious. In the first case, due to the rapid response to IV therapy, a step-down approach to oral TMP-SMX was successfully performed in the last week of treatment, followed by prolonged secondary prophylaxis.

Under normal conditions, the response to therapy should be assessed after at least 8 days of treatment, primarily based on clinical improvement of respiratory function. Due to conflicting data available in the literature, assessing the therapeutic response by monitoring serum levels of BDG or qPCR in respiratory samples is not actually recommended for assessing therapeutic response due to conflicting data [103]. Nevertheless, a recent retrospective trial suggested that PCR conversion is related to prognosis. Thus, a positive qPCR result can be considered as an indicator of treatment failure [104]. The evaluation of treatment failure should also include careful evaluation of hospital-acquired infections and non-infectious secondary complications (such as pneumothorax or pleural effusion), which may be mistakenly interpreted as a lack of response to therapy.

In the case of confirmed treatment failure with first-line therapy, the use of salvage rescue therapy is indicated [105]. The combination of primaquine + clindamycin represents the first choice in this therapeutic alternative since IV pentamidine, another potentially effective therapeutic alternative, is also a valid option but in some observational studies has been associated with higher mortality, both in HIV and non-HIV patients [100,106]. Finally, the use of the combination of TMP-SMX plus an echinocandin, usually caspofungin, has been described as possible salvage therapy, but the evidence is supported by limited evidence, including solely case reports [104,107], and animal models only [108].

Finally, the use of the combination of TMP-SMX plus an echinocandin, usually caspofungin, has been described as possible salvage therapy, but the evidence is supported by limited evidence, including case reports [104,107], and animal models [108]. Recent retrospective studies showed how the use of TMP-SMX in combination with caspofungin as a first-line therapy in non-HIV patients with severe disease seems to be associated with a better outcome, with no increase in adverse effects [109,110]. However, in these studies, little to no benefit seems to be associated with the use of TTMP-SMX plus caspofungin as a second-line therapy [109]. These findings confirm that failure to respond to first-line therapy is strongly linked to a worse prognosis, and evidences supporting any salvage therapy strategy over others are lacking [109].

The second case report described a clinical worsening after a full week of adequate treatment. Considering risk factors for secondary complications, such as hospital-acquired infection, broad-spectrum antimicrobials were initiated, including an echinocandin. However, a possible lack of response to first-line Pj therapy was considered.

Systemic corticosteroid therapy is often recommended in PcP, particularly in patients with respiratory failure. The evidence for corticosteroid therapy mainly comes from HIV patients, and while data for non-HIV patients are inconclusive [111,112,113]. A recent systematic review and meta-analysis, including 11 retrospective studies, indicates that treatment with steroids in non-HIV PcP patients is associated with a worse outcome, with a 29% increased overall risk of death [114]. Therefore, using corticosteroids in the HIV-negative population requires careful consideration. Both presented cases were treated with systemic corticosteroids. Concerns regarding the development of secondary pulmonary fibrosis had certainly driven this approach, although the current literature does not support this strategy, which accordingly should be avoided.

Among the adjunctive therapies, the identification and treatment of any viral co-infections are often mentioned for the treatment of PcP. In particular, the diagnosis and treatment of a possible CMV co-infection are frequently described [47]. However, it is important not to underestimate other viral infections or any healthcare-associated infections. In the second case presented, a viral respiratory illness was correctly diagnosed and treated.

Lastly, in SOT recipients with severe infectious complications, reducing the intensity of immunosuppression is often taken into consideration. From a pathophysiological point of view, the immune response plays a fundamental role in the healing of infections. Evidence regarding reducing immunosuppressive therapy in PcP is even less. A retrospective study including 20 patients with PcP in SOT recipients demonstrated that the surviving patients showed an earlier reduction in immunosuppressive therapy [115]. The available data, even in the context of bacterial infections, suggest that this strategy is not associated with a significant increase in the risk of rejection [116]. In the absence of definite evidence, most guidelines suggest considering a reduction in the immunosuppressive regimen. The most common strategy usually involves discontinuing anti-proliferative agents such as MMF and MPA, which have the therapeutic effect of reducing the proliferation of lymphocytes and thus total lymphocyte count. As a second option, it is recommended to reduce the dosage of calcineurin inhibitors to the lowest tolerated level. However, the impact of such strategies on patient outcomes is not supported by any scientific evidence [117].

In both cases presented, immune-suppressive therapy was reduced once PcP was diagnosed. As per local protocol for severe infections, MMF was discontinued in both patients, and the dosage of calcineurin inhibitors was reduced. Further studies are certainly needed to assess the effectiveness of these strategies, not only in PcP but in all major infections in highly immune-suppressed hosts.

## 5. Conclusions and Future Perspectives

The management of PcP in heart transplant recipients requires a comprehensive and individualized approach to minimize the risk of infection and optimize patient outcomes. As evidenced by the cases discussed, the timing of the infection’s presentation can sometimes be unpredictable, occurring many years after the so-called “risk period”. Current epidemiological data might underestimate the PcP risk in this population.

While universal prophylaxis is recommended in the first 6 to 12 months post-transplantation, opportunistic infections can occur beyond this timeframe. The literature is not consistent on the best prevention strategy to adopt beyond the first year post-transplantation. While lifelong prophylaxis may be considered excessive for most heart transplant recipients, a thorough assessment of individual risk factors should be regularly performed to optimize prevention strategies. Careful consideration should be given to temporarily acquired predisposing factors, such as intercurrent viral diseases and changes in immunosuppressive therapies. Thorough and careful follow-up evaluations by the transplant team should be considered to appropriately tailor prophylaxis in high-risk patients. This should include unconventional and poorly understood risk factors, such as viral illnesses other than CMV or new immunosuppressive therapies. These factors, despite lacking formal indications for prophylaxis, should be carefully evaluated in assessing the risk in each individual patient.

The possibility of PcP should always be considered in the evaluation of a heart transplant recipient presenting with acute respiratory failure. Moreover, PcP should also be considered in the differential diagnosis of any type of solid organ transplant recipient, especially in the absence of universal prophylaxis. Clinical suspicion remains a major factor in reaching a timely diagnosis, avoiding diagnostic and therapeutic delays that could affect the course and prognosis. During the diagnostic evaluation, heightened vigilance should focus on assessing risk factors, such as the lack of Pc prophylaxis, previous PcP history, viral infections (particularly CMV), and the presence of high-dose immunosuppressive therapy. An early and targeted diagnosis using non-invasive markers such as serum BDG and specific tests like qPCR is essential for timely initiation of therapy. High-dose TMP-SMX remains the preferred first-line treatment and should be promptly initiated. Finally, adjunctive therapies such as the treatment of viral co-infections, judicious use of corticosteroids, and temporary reduction in immunosuppressive therapy should be considered on an individual basis.

It is crucial to individualize the management approach based on the patient’s characteristics and clinical judgment. Further research is needed to establish robust evidence-based recommendations for the prevention and treatment of PcP in heart transplant recipients.

## Figures and Tables

**Figure 1 pathogens-12-01265-f001:**
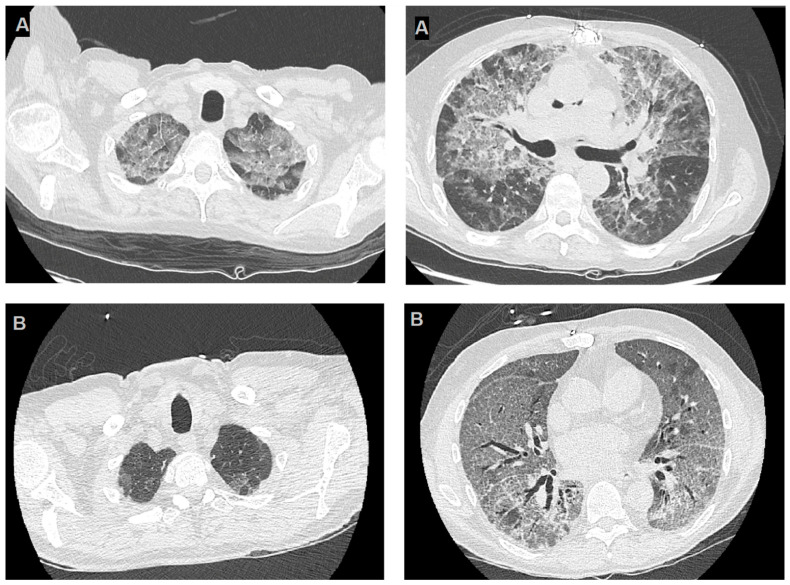
High-resolution computerized tomography images of cases reported. Panels (**A**) highlight two slices from HRCT of the first reported case, the images show bilateral diffuse GGOs, with apical predominance and peripheral sparing and panels (**B**) refer to the second case and show diffuse thin GGOs in both lungs, without apical predominance or peripheral sparing.

**Figure 2 pathogens-12-01265-f002:**
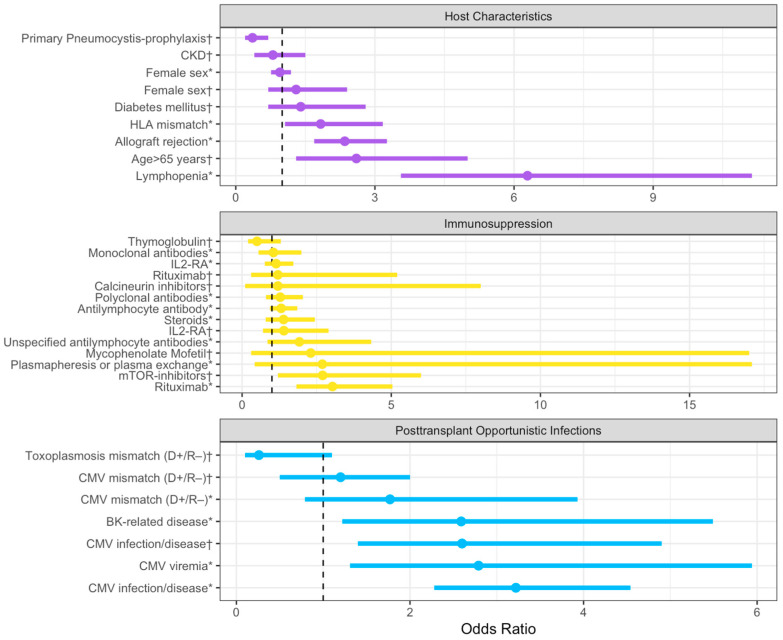
Summary of odds ratios with corresponding 95% confidence intervals extracted from the literature, categorized based on host characteristics, immunosuppression, and post-transplant infections. The dashed line represents an OR value of 1. The plot was realized with Rstudio, RStudio Team (2020). Abbreviations: CKD: chronic kidney disease; HLA: Human leukocyte antigen; IL-RA: IL-2 receptor antagonist; and CMV: Cytomegalovirus. (*): Data from Permpalung, N. et al. [12]; (†): Data from Neofytos, D. et al. [6].

**Figure 3 pathogens-12-01265-f003:**
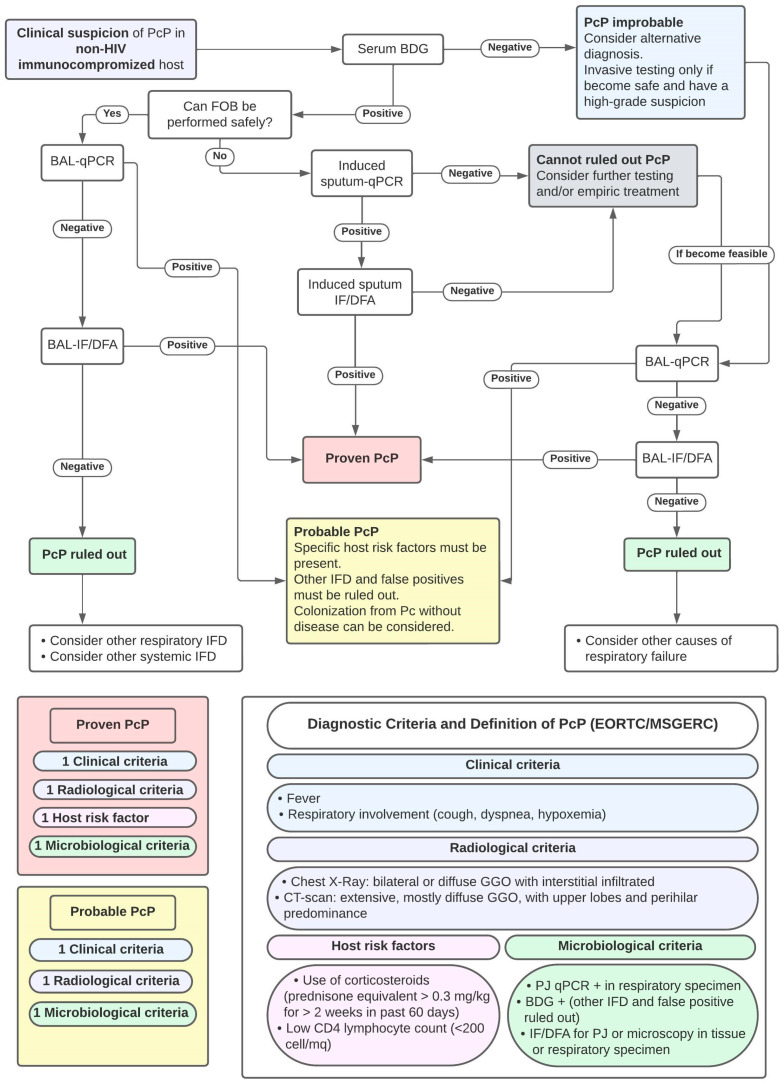
Proposal of Diagnostic Approach to PcP diagnosis and Criteria for PcP Diagnosis. Abbreviations: FOB: fiber-optic bronchoscopy; BDG: Beta-D-Glucan; BAL: broncho-alveolar lavage; qPCR: quantitative polymerase chain reaction for PJ; IF/DFA: immunofluorescence/direct immunofluorescent antibodies; DD: differential diagnosis; and IFD: invasive fungal diseases.

**Table 1 pathogens-12-01265-t001:** Baseline maintenance immuno-suppression regimens, admission’s laboratory values and imaging of reported cases.

Laboratory Values:	Case 1	Case 2
WBC:	4.79 × 10^9^/L	13.2 × 10^9^/L
Lymphocytes:	340 cells/mL	530 cells/mL
CRP:	220.1 mg/L	23.2 mg/L
PCT:	0.56 mcg/L	0.02 mcg/L
LDH:	541 UI/L (+4)	315 UI/L
BDG:	264.1 pg/mL (+5)	207.7 pg/mL (+3)
Imaging:	Case 1	Case 2
Chest X-ray:	Diffuse alveolar-interstitial opacities, possible pulmonary oedema	Diffuse bilateral thickenings
RCT:	Multiple symmetrical ground-glass opacities. Apical and perihilar prevalence.	Ubiquitous ground glass thickenings. Anterior–posterior gradient.
Immuno-Suppresion Medications	Case 1	Case 2
Mofetil Mycophenolate	1000 mg BID	1500 mg BID
Ciclosporin	125 mg BID	100 mg + 75 mg daily
Prednisone	10 mg daily	-

Abbreviations: WBC, white blood cells; CPR, C-Reactive protein; PCT, Procalcitonin; LDH, Lactic De-Hydrogenase; BDG, Beta-D-Glucan; HRCT, high-resolution computerized tomography; and BID: twice a day.

**Table 2 pathogens-12-01265-t002:** Universal, Expanded and Proposed Criteria for PcP Prophylaxis among the current literature.

Current Recommendations on PcP Prophylaxis	
Universal prophylaxis is recommended for at least 6–12 months following heart transplant.	Class I, Level of Evidence B ISHLT Guidelines for the Care of Heart Transplant Recipients
Extended prophylaxis can be considered: chronic CMV infection; prior PcP; those who require augmented immunosuppression.	Class IIa, Level of Evidence B ISHLT Guidelines for the Care of Heart Transplant Recipients
Prolonged prophylaxis is indicated in SOT recipients who show: Increase immunotherapy for graft rejection; CMV infection; Corticosteroids (>20 mg/die PEQ); Prolonged neutropenia; Flares of autoimmune disease.	N.A. American Society of Transplantation Infectious Diseases Community of Practice
Lifelong prophylaxis in SOT patients with history of PcP	N.A.
	American Society of Transplantation Infectious Diseases Community of Practice
Proposed Recommendations on PcP Prophylaxis	
Extended prophylaxis: Allograft rejection; Detectable CMV viremia.	Hosseini-Moghaddam et al. [8]
Extended prophylaxis: Lymphopenia; BK polyomavirus-related infections; rituximab exposure.	Permpalung et al. [12]
Prolonged prophylaxis: Corticosteroids > 10 mg/die and age > 65 yrs; Corticosteroids > 10 mg/die and other immuno-suppressive agent.	Malpica et al. [35]
Lifelong prophylaxis in all HT recipients	Fillatre et al. [32]

Abbreviations: ISHLT, International Society for Heart and Lung Transplantation; CMV, Cytomegalovirus; SOT, solid organ transplantation; PEQ, prednisone equivalents; HT, heart transplantation.

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
