# Peer review of "Pneumocystis jirovecii Pneumonia after Heart Transplantation: Two Case Reports and a Review of the Literature"

_pathogens, 2023, doi:10.3390/pathogens12101265_

Round 1

Reviewer 1 Report

The article written by Burzio et al is interesting and well performed. However, the authors need to make some recommendations before publishing the text.

 Title.  It must be changed; your experience is not for two cases. I recommend, Pneumocystis jirovecii Pneumonia after Heart Transplantation: two case reports and review of the literature.

 Abstract. The term PjP is not correct and should be changed to PcP. Please revise: Nevez G, Totet A, Matos O, Calderon EJ, Miller RF, Le Gal S. It is still PCP that can stand for Pneumocystis pneumonia: Appeal for generalized use of only one acronym. Med Mycol. 2021 Jul 14;59(8):842-844. doi: 10.1093/mmy/myab024. PMID: 34003930.

The term re-emerging disease is not correct, Please rewrite this idea.

 Keywords. Delete re emerging opportunistic infections.

 Introduction

The authors must rewrite this sentence: Pneumocystis jirovecii pneumonia (PJP) is a rare opportunistic infection. There is increasing evidence of infection by Pneumocystis jirovecii

Clinical Cases

Line 79. Delete these features are typical of PJP[14]. This idea is shown in the text.

Line 80. Delete without any findings specific for PJP; both 80 imaging can be associated with PJP[15]. This idea is shown in the text.

Line 87. The authors must show the type of PCR performed, its target and the reference used.

Line 90-91. The authors must clarify the dose of mycophenolate and steroid therapy.

Line 107. Reference for Beta-D-Glucan technique.

Line 284. Add the reference: Aggoun D, Bleibtreu A, Desiré E, Lecuyer L, Leprince P, Varnous S, Coutance G, Lescroart M; PNC HTX STUDY GROUP. Pneumocystis prophylaxis in French heart transplant centers: A nationwide survey. Transpl Infect Dis. 2023 Jun;25(3):e14053. doi: 10.1111/tid.14053. Epub 2023 Mar 7. PMID: 36882963. Please clarify this idea in the text.

Line 285. Complete the title of Table 2.

Line 339. Add the reference:  Esteves F, Calé SS, Badura R, de Boer MG, Maltez F, Calderón EJ, van der Reijden TJ, Márquez-Martín E, Antunes F, Matos O. Diagnosis of Pneumocystis pneumonia: evaluation of four serologic biomarkers. Clin Microbiol Infect. 2015 Apr;21(4):379.e1-10. doi: 10.1016/j.cmi.2014.11.025. Epub 2014 Dec 4. PMID: 25630458. Please clarify this idea in the text.

Line 363. Add the reference: Bigot J, Vellaissamy S, Senghor Y, Hennequin C, Guitard J. Usefulness of ß-d-Glucan Assay for the First-Line Diagnosis of Pneumocystis Pneumonia and for Discriminating between Pneumocystis Colonization and Pneumocystis Pneumonia. J Fungi (Basel). 2022 Jun 24;8(7):663. doi: 10.3390/jof8070663. PMID: 35887420; PMCID: PMC9318034. Please clarify this idea in the text.

Line 374. The sentence must rewrite. Recently, Riebold D et al (Mahnkopf M, Wicht K, Zubiria-Barrera C, Heise J, Frank M, Misch D, Bauer T, Stocker H, Slevogt H. Axenic Long-Term Cultivation of Pneumocystis jirovecii. J Fungi (Basel). 2023 Sep 1;9(9):903. doi: 10.3390/jof9090903. PMID: 37755011; PMCID: PMC10533121.) obtained Pneumocystis in the axenic medium.

Line 463-463. This sentece must rewrite. Please add two reference and clarify this idea in the text:

 Qi H, Dong D, Liu N, Xu Y, Qi M, Gu Q. Efficacy of initial caspofungin plus trimethoprim/sulfamethoxazole for severe PCP in patients without human immunodeficiency virus infection. BMC Infect Dis. 2023 Jun 16;23(1):409. doi: 10.1186/s12879-023-08372-z. PMID: 37328748; PMCID: PMC10273704.

Wu HH, Fang SY, Chen YX, Feng LF. Treatment of Pneumocystis jirovecii pneumonia in non-human immunodeficiency virus-infected patients using a combination of trimethoprim-sulfamethoxazole and caspofungin. World J Clin Cases. 2022 Mar 26;10(9):2743-2750. doi: 10.12998/wjcc.v10.i9.2743. PMID: 35434110; PMCID: PMC8968794.

Line 591. Reference 16 should be revised

Minor editing of English language required

Author Response

The article written by Burzio et al is interesting and well performed. However, the authors need to make some recommendations before publishing the text.

 Title.  It must be changed; your experience is not for two cases. I recommend, Pneumocystis jirovecii Pneumonia after Heart Transplantation: two case reports and review of the literature.

*****RESPONSE: We appreciate the Reviewer's comment. And we are grateful for the positive comments and the opportunity to revise our paper following the Reviewer’s advice. We acknowledge that the use of the term "our experience" in the title, while impactful, may not entirely correspond to the content of the article itself. The description containing "Two case reports and review..." is certainly more accurate.

 Abstract. The term PjP is not correct and should be changed to PcP. Please revise: Nevez G, Totet A, Matos O, Calderon EJ, Miller RF, Le Gal S. It is still PCP that can stand for Pneumocystis pneumonia: Appeal for generalized use of only one acronym. Med Mycol. 2021 Jul 14;59(8):842-844. doi: 10.1093/mmy/myab024. PMID: 34003930.

*****RESPONSE: We appreciate the Reviewer’s suggestion.  It is a difficult issue to get around.   We are grateful for the opportunity to clarify. We were aware of the mentioned article, which advocates the use of the acronym PcP to refer to Pneumocystis Pneumonia instead of the acronym PJP. We would like to emphasize that the cited article represents the opinion of its authors, and to date, there are hundreds of publications that use the acronym PJP. The article has been modified to use the acronym PcP; furthermore, a sentence has been added to the introduction to indicate that the acronym PJP is also widely used, citing the reference you brought to our attention:

"This infection is referred to as PcP (Pneumocystis Pneumonia), but the abbreviation PJP (referring to the renaming of the pathogen as Pneumocystis Jirovecii) is also commonly used, although formally incorrect."

The term re-emerging disease is not correct, Please rewrite this idea.

 Keywords. Delete re emerging opportunistic infections.

The authors must rewrite this sentence: Pneumocystis jirovecii pneumonia (PJP) is a rare opportunistic infection. There is increasing evidence of infection by Pneumocystis jirovecii

*****RESPONSE:  We thank the Reviewer for the assessment and perspectives The idea that Pneumocystis pneumonia (PcP) is a clinically relevant infection and not an unusual occurrence is a fundamental part of our article. Additionally, we would like to emphasize that the concept of PcP as a "re-emerging disease" has been used by several authors before us, while technically incorrect. The two parts have been modified as suggested to further highlight these concepts, as follows:

"Post-transplant Pneumocystis jirovecii Pneumonia (PcP) is an uncommon but increasingly reported disease among Solid Organ Transplantation (SOT) recipients, associated with significant morbidity and mortality."

"Pneumocystis jirovecii pneumonia (PcP) is a disease initially identified as a rare opportunistic infection occurring in HIV patients [1]. In subsequent years, the role of Pneumocystis jirovecii as a human pathogen in many circumstances, including immunocompromised non-HIV patients, has been established [4]."

Line 79. Delete these features are typical of PJP[14]. This idea is shown in the text.

Line 80. Delete without any findings specific for PJP; both 80 imaging can be associated with PJP[15]. This idea is shown in the text.

These changes have been made as requested

Line 87. The authors must show the type of PCR performed, its target and the reference used.

*****RESPONSE: We are grateful for the opportunity to clarify. The following lines were added:

"For rapid Pneumocystis jirovecii detection, eazyplex Pneumocystis jirovecii kit on Amplex Genie II platform (AmplexDiagnostics GmbH, Germany) were used directly from bronchoalveolar lavage specimen."

To better clarify this kind of method, a section was added to diagnostic test paragraph to explain LAMP tests:

Several new methods to diagnose PcP have been proposed, including antibiodies assays, new targets for PCR, and Loop Mediated Isothermal Amplification (LAMP).[70] LAMP provide an alternative to PCR methods, as it can amplify a target gene with only an heating device and isothermal conditions. [cit1] In both HIV and non-HIV populations, LAMP had demonstrated high sensivity (84%-99%) and specificity (96%-99%), comparable to PCR methods. [70, cit1, cit2, cit3] However, in one study LAMP quantification method, know as Time-To-Positivity (TTP), showed a worse correlation to fungal load than Cycle-Threshold (Ct) of qPCR, that still remain the gold standard.[cit3]

New citations are the following:

Singh P, Singh S, Mirdha BR, Guleria R, Agarwal SK, Mohan A. Evaluation of Loop-Mediated Isothermal Amplification Assay for the Detection of Pneumocystis jirovecii in Immunocompromised Patients. Mol Biol Int. 2015;2015:819091. doi: 10.1155/2015/819091. Epub 2015 Nov 19. PMID: 26664746; PMCID: PMC4668309.

Huber T, Serr A, Geißdörfer W, Hess C, Lynker-Aßmus C, von Loewenich FD, Bogdan C, Held J. Evaluation of the Amplex eazyplex Loop-Mediated Isothermal Amplification Assay for Rapid Diagnosis of Pneumocystis jirovecii Pneumonia. J Clin Microbiol. 2020 Nov 18;58(12):e01739-20. doi: 10.1128/JCM.01739-20. PMID: 32938732; PMCID: PMC7685886.

Scharmann U, Kirchhoff L, Schmidt D, Buer J, Steinmann J, Rath PM. Evaluation of a commercial Loop-mediated Isothermal Amplification (LAMP) assay for rapid detection of Pneumocystis jirovecii. Mycoses. 2020 Oct;63(10):1107-1114. doi: 10.1111/myc.13152. Epub 2020 Aug 12. PMID: 32738076.

 Line 90-91. The authors must clarify the dose of mycophenolate and steroid therapy.

*****RESPONSE:  We are grateful for the opportunity to clarify.  These changes have been made as requested. The type and dosage of the immunosuppressive regimen have been included in Table 1. Consequently, the table title has been modified as follows:

Table 1. Maintenance Immuno-Suppression Regimen before Admission, Admission’s Laboratory Values, and Imaging of Reported Cases.

Lines included in the table are the following:

Immuno-Suppresion Medications

Case 1

Case 2

Mofetil Mycophenolate

1000mg BID

1500mg BID

Ciclosporin

125mg BID

100mg + 75mg daily

Prednisone

10mg daily

/

The changes in the steroid therapy for Case 1 have been added as follows:

"… modified by discontinuing mycophenolate (stopping for the first week, then 250mg BID) and increasing steroid therapy (switch to IV Methyl-Prednisone, 125mg TID)."

Line 107. Reference for Beta-D-Glucan technique.

*****RESPONSE:  We are grateful for the opportunity to clarify.  These changes have been made as requested :

“Quantitative serum Serum Beta-D-Glucan (BDG) was performed (ß-Glucan Test on Toxinometer MT-6500, FUJIFILM Wako Pure Chemical Corporation, Japan) and resulted positive (207.7 pg/dl, reference value <7pg/dl)”

Line 284. Add the reference: Aggoun D, Bleibtreu A, Desiré E, Lecuyer L, Leprince P, Varnous S, Coutance G, Lescroart M; PNC HTX STUDY GROUP. Pneumocystis prophylaxis in French heart transplant centers: A nationwide survey. Transpl Infect Dis. 2023 Jun;25(3):e14053. doi: 10.1111/tid.14053. Epub 2023 Mar 7. PMID: 36882963. Please clarify this idea in the text.

*****RESPONSE:  We are grateful for the opportunity to clarify.  The proposed change has been integrated into the text as follows:

"Nevertheless, there is considerable variability in the application of these recommendations. A recent French survey has demonstrated that, even within the same national transplant network, there is a significant variability in PcP prophylaxis, particularly concerning the dosage used and the duration of therapy."

Line 285. Complete the title of Table 2.

*****RESPONSE:  We thank the reviewer for the advice. We completed the title of the table as the reviewer asked.

Line 339. Add the reference:  Esteves F, Calé SS, Badura R, de Boer MG, Maltez F, Calderón EJ, van der Reijden TJ, Márquez-Martín E, Antunes F, Matos O. Diagnosis of Pneumocystis pneumonia: evaluation of four serologic biomarkers. Clin Microbiol Infect. 2015 Apr;21(4):379.e1-10. doi: 10.1016/j.cmi.2014.11.025. Epub 2014 Dec 4. PMID: 25630458. Please clarify this idea in the text.

*****RESPONSE:  We are grateful for the opportunity to clarify.  The proposed change has been integrated into the text as follows:

"In a retrospective study, the two markers previously described (LDH and BDG) were compared with two other serological markers for the diagnosis of PcP, namely Krebs von den Lungen 6 antigen (KL-6) and S-Adenosyl-Methionine (SAM). KL-6 is a host molecule, strongly expressed on type II alveolar pneumocytes and bronchiolar epithelial cells; KL-6 was linked to interstitial lung disease rather than being a specific marker of PcP. SAM is an endogenous metabolic intermediate that is required in the metabolism of at least some strains of Pneumocystis. In this study, BDG was found to be the most reliable serologic biomarker for PcP diagnosis. Interestingly, in this study, the combination of BDG and KL-6 was the most accurate serologic approach (94.3% sensitivity and 89.6% specificity) and was therefore proposed as a minimally invasive diagnostic approach."

Line 363. Add the reference: Bigot J, Vellaissamy S, Senghor Y, Hennequin C, Guitard J. Usefulness of ß-d-Glucan Assay for the First-Line Diagnosis of Pneumocystis Pneumonia and for Discriminating between Pneumocystis Colonization and Pneumocystis Pneumonia. J Fungi (Basel). 2022 Jun 24;8(7):663. doi: 10.3390/jof8070663. PMID: 35887420; PMCID: PMC9318034. Please clarify this idea in the text.

*****RESPONSE:  We are grateful for the opportunity to clarify.  The proposed change has been integrated into the text as follows:

“Furthermore, in combination with molecular methods (see below), BDG assay can be used to quantify the pathogen load and therefore confidently distinguish between infected and colonized patients. [cit]“

Line 374. The sentence must rewrite. Recently, Riebold D et al (Mahnkopf M, Wicht K, Zubiria-Barrera C, Heise J, Frank M, Misch D, Bauer T, Stocker H, Slevogt H. Axenic Long-Term Cultivation of Pneumocystis jirovecii. J Fungi (Basel). 2023 Sep 1;9(9):903. doi: 10.3390/jof9090903. PMID: 37755011; PMCID: PMC10533121.) obtained Pneumocystis in the axenic medium.

*****RESPONSE:  We are grateful for the opportunity to clarify.  The proposed change has been integrated into the text as follows:

“Traditionally, microbiological diagnosis of PJP is considered challenging due to the inability to culture the microorganism in standard culture media[70]. Pneumocystis jirovecii can grow in vitro on selected terrains, but these systems are complex, expensive, and not useful for routine use.[71–73] Only very recently, a stable PJ culture was developed, using an axemic medium system; albeit further optimization of the culture conditions is needed, this approach is promising to obtain PJ cultures for clinical purposes. [cit]”

Line 463-463. This sentece must rewrite. Please add two reference and clarify this idea in the text:

 Qi H, Dong D, Liu N, Xu Y, Qi M, Gu Q. Efficacy of initial caspofungin plus trimethoprim/sulfamethoxazole for severe PCP in patients without human immunodeficiency virus infection. BMC Infect Dis. 2023 Jun 16;23(1):409. doi: 10.1186/s12879-023-08372-z. PMID: 37328748; PMCID: PMC10273704.

Wu HH, Fang SY, Chen YX, Feng LF. Treatment of Pneumocystis jirovecii pneumonia in non-human immunodeficiency virus-infected patients using a combination of trimethoprim-sulfamethoxazole and caspofungin. World J Clin Cases. 2022 Mar 26;10(9):2743-2750. doi: 10.12998/wjcc.v10.i9.2743. PMID: 35434110; PMCID: PMC8968794.

*****RESPONSE:  We are grateful for the opportunity to clarify.  The proposed change has been integrated into the text as follows:

“Finally, the use of the combination of TMP-SMX plus an echinocandin, usually  caspofungin, has been described as possible salvage therapy, but the evidence is supported by limited evidence, including case reports [95,98], and animal models [99]. Recent retrospective studies showed how the use of TMP-SMX in combination with caspofungin as a first-line therapy in non-HIV patients with severe disease seems to be associated to better outcome, with no increase of adverse effects. [cit]1-2] However, in these studies, little to no benefit seems associated to the use of TTMP-SMX plus caspofungin as a second-line therapy. [cit1] These findings confirm that failure to respond to first-line therapy is strongly linked to a worse prognosis, and evidences supporting any salvage therapy strategy over others are lacking [cit1].”

Line 591. Reference 16 should be revised

*****RESPONSE:  We are grateful for the opportunity to clarify.  We have revised that.

Reviewer 2 Report

In the review article "Pneumocystis jirovecii Pneumonia after Heart Transplantation: 2 our experience and evidence from the literature", the authors summarized the current timing of PJP prophylaxis, the specific potential risk factors for PJP after HT, and the determinants of a prompt diagnosis and therapeutic approach in critically ill patients. The review is very well structured and well written. The topic itself is medically important and relevant in today's time where a plethora of factors can influence the host-pathogen interactions and the disease outcome. There is a minor error in the title which needs to be corrected. The word "form literature" needs to be changed to "from literature"

Thanks 

Author Response

*****RESPONSE:  We thank the Reviewer for the supportive comments!  The article title has been changed, following the recommendations of another reviewer, to "Pneumocystis jirovecii Pneumonia after Heart Transplantation: Two Case Reports and a Review of the Literature”

Reviewer 3 Report

The manuscript from Burzio, et al., is a nice literature review that mixes in some data from the authors detailing their experience with Pneumocystis infection in patients after heart transplant.  The paper and review are well written.  The only comment I have has to do with the failure to mention CD4 T cell levels in the patient cases shown.  There is not much discussion about CD4 T cells at all and even if they are not predictive of PJP in solid organ transplant patients, this should at least be commented on. 

Author Response

*****RESPONSE:  We thank the Reviewer for the supportive comments!   CD4 levels were not measured upon admission for these patients, nor in the immediate days following admission, but only after several days of hospitalization. Both patients were not immediately admitted to the ICU upon admission. The obtained values were not mentioned as they were considered less relevant due to the timing of their determination.

To emphasize the importance of the role of CD4 in these patients, additional text has been added to paragraph 3.2:

"Interestingly, beyond lymphocytopenia, which is one of the most studied and recognized risk factors for PcP, the role of CD4 lymphocytes in the pathogenesis of non-HIV related PcP has recently been acknowledged [cit1-2]. One retrospective study had demonstrated that a low level of CD4 lymphocytes is correlated with the development of PcP. [cit1] Furthermore, in two retrospective studies involving non-HIV PcP patients, low CD4 lymphocyte levels appear to be associated with a worse prognosis [cit1-2]."

Cao Y, Chen J, Dong L. Supplementary Role of Immunological Indicators in the Diagnosis and Prognosis of Pneumocystis Pneumonia in Non-HIV Immunocompromised Patients. Infect Drug Resist. 2022 Aug 21;15:4675-4683. doi: 10.2147/IDR.S372427. PMID: 36034170; PMCID: PMC9406888.

Duan J, Gao J, Liu Q, Sun M, Liu Y, Tan Y, Xing L. Characteristics and Prognostic Factors of Non-HIV Immunocompromised Patients With Pneumocystis Pneumonia Diagnosed by Metagenomics Next-Generation Sequencing. Front Med (Lausanne). 2022 Mar 3;9:812698. doi: 10.3389/fmed.2022.812698. PMID: 35308503; PMCID: PMC8928194.

Reviewer 4 Report

Dear authors

The cases presented in the article are interesting. I have a few considerations about it: 

In both presented cases it was possible to obtain BAL samples. Why did not you submit those samples to mycological direct exams with easy to perform stains (Giemsa, Gram-Weigher, Grocott, calcofluor)? Those simple preparations could have render good and quick results. Besides, you can also use those samples to search for PJ-DNA, and even you could have studied if there were any DPHS gene mutations. Different stains are much more specific than BDG (it is a panfungal biomarker), are cheaper, easier, and fast.

1.       In your work there is no information about microscopic diagnosis using staining methods or direct immunofluorescence. Those methodologies are very useful, their sensitivity varies according to the material, laboratory experience, type of patients, is less sensitive in HIV-negative but is more specific than B-D-glucan and are available and affordable in most settings.

Please see:  

a. Salzer HJF et al. Clinical, diagnostic, and treatment disparities between HIV-infected and non-HIV-infected immunocompromised patients with Pneumocystis jirovecii pneumonia. Respiration, DOI:10.1159/000487713.

b. Bateman M, et al. Diagnosing Pneumocystis jirovecii pneumonia: a review of current methods and novel approaches. Med. Mycol. 2020, 58: 1015-28. (ref 70).

c. Kato H, et al. Diagnosis and treatment of Pneumocystis jirovecii pneumonia in HIV-infected or non-HIV-infected patients –difficulties in diagnosis and adverse effects of trimethoprim-sulfamethoxazole. J Infect Chemother 2019; 25: 920-4.

d. Senécal J et al. Non-invasive diagnosis of Pneumocystis jirovecii pneumonia: a systematic review and meta-analysis. Clin Microbiol Infect 2022; 28: 23-30.

Even in the guidelines from the American Society of transplantation infectious diseases community of practice (ref. 47) in table 3 are the recommended diagnostic techniques for the investigation of PjP in different samples, there it is showed that stains  (silver, calcofluor, or others) have a strong level of recommendation and high quality of evidence; on the other hand B-D-glucan has weak levels of recommendation and low quality of evidence.

On the other hand, you should consider that BDG is not available in many countries.

Why did you choose using caspofungina associated with TMP-SMX instead of other antifungal drugs. The results with caspofungin are not good enough.

Numerous studies have indicated that sulfa prophylaxis failure and poor treatment outcomes are associated with DHPS gene mutations in the target organism. Mutations associated with sulfa resistance have been observed in the DHPS gene at codons 55 and 57. Did you study if there were any of these mutations in your cases?

In table 1 there are some figures with decimals after a comma not after a dot.

In line 405: you mention the criteria for the definition of PJP proposed by EORTC/MSGERC, but the corresponding reference is not 86.

Author Response

The cases presented in the article are interesting. I have a few considerations about it: 

In both presented cases it was possible to obtain BAL samples. Why did not you submit those samples to mycological direct exams with easy to perform stains (Giemsa, Gram-Weigher, Grocott, calcofluor)? Those simple preparations could have render good and quick results. Besides, you can also use those samples to search for PJ-DNA, and even you could have studied if there were any DPHS gene mutations. Different stains are much more specific than BDG (it is a panfungal biomarker), are cheaper, easier, and fast.

  1. In your work there is no information about microscopic diagnosis using staining methods or direct immunofluorescence. Those methodologies are very useful, their sensitivity varies according to the material, laboratory experience, type of patients, is less sensitive in HIV-negative but is more specific than B-D-glucan and are available and affordable in most settings.

Please see:  

  1. Salzer HJF et al. Clinical, diagnostic, and treatment disparities between HIV-infected and non-HIV-infected immunocompromised patients with Pneumocystis jirovecii pneumonia. Respiration, DOI:10.1159/000487713.
  2. Bateman M, et al. Diagnosing Pneumocystis jirovecii pneumonia: a review of current methods and novel approaches. Med. Mycol. 2020, 58: 1015-28. (ref 70).
  3. Kato H, et al. Diagnosis and treatment of Pneumocystis jirovecii pneumonia in HIV-infected or non-HIV-infected patients –difficulties in diagnosis and adverse effects of trimethoprim-sulfamethoxazole. J Infect Chemother 2019; 25: 920-4.
  4. Senécal J et al. Non-invasive diagnosis of Pneumocystis jirovecii pneumonia: a systematic review and meta-analysis. Clin Microbiol Infect 2022; 28: 23-30.

Even in the guidelines from the American Society of transplantation infectious diseases community of practice (ref. 47) in table 3 are the recommended diagnostic techniques for the investigation of PjP in different samples, there it is showed that stains  (silver, calcofluor, or others) have a strong level of recommendation and high quality of evidence; on the other hand B-D-glucan has weak levels of recommendation and low quality of evidence.

On the other hand, you should consider that BDG is not available in many countries.

*****RESPONSE:  We appreciate the Reviewer’s comment and perspective. We fully agree that the role of traditional diagnostics is supported by robust literature, as the reviewer suggests, and that the use of stains is more cost-effective than biomarkers and PCR-based methods. However, this does not automatically translate into being easier or faster. In many laboratories, including ours, the use of automated tests is becoming increasingly important to optimize workflow. Requesting one or more specific stains would have taken days in our laboratory, or would have required sending samples to an external laboratory, leading to increased costs and time. The limited use of traditional diagnostics also reduces its effectiveness, as you rightly point out. At the same time, BDG, while indeed less specific and more expensive, is a widely used automated test (at least in our part of the world) that provides rapid results in a short time. However, we realize that these considerations need to be tailored to the specific clinical situation.

To emphasize the importance of traditional diagnostics, the following modification has been made to the text:

"Traditionally, microbiological diagnosis of PJP is considered challenging due to the inability to culture the microorganism in standard culture media [70]. Pneumocystis jirovecii can grow in vitro on selected terrains, but these systems are complex, expensive, and not useful for routine use. [71–73] Only very recently, a stable PJ culture was developed, using an axenic medium system; albeit further optimization of the culture conditions is needed, this approach is promising to obtain PJ cultures for clinical purposes [cit].

Classically, PcP diagnosis was confirmed via direct visualization of the pathogen via staining. [70]  While these tests are easy and cheap to perform, they lack sensitivity due to dependence on the quality of the sample and observer interpretation, particularly when fungal burden is low, as in non-HIV patients. [70] However, they maintain a high-grade recommendation in most guidelines, due to robust supporting literature (albeit most in HIV population). [47].

Immunofluorescence, introduced later and using monoclonal antibodies to Pneumocystis jirovecii, also known as direct immunofluorescent antibodies (DFA), appears to have greater sensitivity[50]. Other methods, using polymerase chain reaction (PCR), appear to have higher sensitivity[70]. The introduction of real-time PCR or quantitative PCR (qPCR) allowed rapid quantitative diagnostic results[71]. Both methods can be performed on sputum and deep samples such as bronchoalveolar lavage (BAL) or transbronchial biopsy, and are becoming increasingly important in the diagnosis of PcP [70]."

Cit:  Mahnkopf M, Wicht K, Zubiria-Barrera C, Heise J, Frank M, Misch D, Bauer T, Stocker H, Slevogt H. Axenic Long-Term Cultivation of Pneumocystis jirovecii. J Fungi (Basel). 2023 Sep 1;9(9):903. doi: 10.3390/jof9090903. PMID: 37755011; PMCID: PMC10533121

Why did you choose using caspofungina associated with TMP-SMX instead of other antifungal drugs. The results with caspofungin are not good enough.

*****RESPONSE:  We are grateful for the opportunity to clarify.  All salvage therapies described in literature show poor results, especially in non-HIV patients. Furthermore, there are increasing evidences, still with very low quality (retrospective studies), about the use of TMP-SMX plus Caspofungin. To further highlight these points, the text was modified as below:

“Finally, the use of the combination of TMP-SMX plus an echinocandin, usually  caspofungin, has been described as possible salvage therapy, but the evidence is supported by limited evidence, including case reports [95,98], and animal models [99]. Recent retrospective studies showed how the use of TMP-SMX in combination with caspofungin as a first-line therapy in non-HIV patients with severe disease seems to be associated to better outcome, with no increase of adverse effects. [cit1-2] However, in these studies, little to no benefit seems associated to the use of TTMP-SMX plus caspofungin as a second-line therapy. [cit1] These findings confirm that failure to respond to first-line therapy is strongly linked to a worse prognosis, and evidences supporting any salvage therapy strategy over others are lacking [cit1].”

Being the two new citations the followings:

  1. Qi H, Dong D, Liu N, Xu Y, Qi M, Gu Q. Efficacy of initial caspofungin plus trimethoprim/sulfamethoxazole for severe PCP in patients without human immunodeficiency virus infection. BMC Infect Dis. 2023 Jun 16;23(1):409. doi: 10.1186/s12879-023-08372-z. PMID: 37328748; PMCID: PMC10273704.
  2. Wu HH, Fang SY, Chen YX, Feng LF. Treatment of Pneumocystis jiroveciipneumonia in non-human immunodeficiency virus-infected patients using a combination of trimethoprim-sulfamethoxazole and caspofungin. World J Clin Cases. 2022 Mar 26;10(9):2743-2750. doi: 10.12998/wjcc.v10.i9.2743. PMID: 35434110; PMCID: PMC8968794.

Please, note that paper number 1 was published after the clinical case we described took place. Considering this evidence, probably we should have treat the patient with TMP-SMX plus Caspofungin as first-line therapy, and change the salvage therapy in case of non-response. However, others second-line therapies are not easily employed in critical ill patients.

Numerous studies have indicated that sulfa prophylaxis failure and poor treatment outcomes are associated with DHPS gene mutations in the target organism. Mutations associated with sulfa resistance have been observed in the DHPS gene at codons 55 and 57. Did you study if there were any of these mutations in your cases?

*****RESPONSE:  We are grateful for the opportunity to clarify.  We did not studied DHPS mutations in our patients. Albeit very interesting, and with upmost clinical impact, this kind of testing is not performed in every laboratory, and can be very difficult to obtain. Furthermore, we believed that a discussion regard the detection of DHPS mutation to assess sulfa-susceptibility, as well others susceptibility testings (like susceptibility to echinocandins, for example), required a chapter itself and exceed the purpose of this review.

In table 1 there are some figures with decimals after a comma not after a dot.

In line 405: you mention the criteria for the definition of PJP proposed by EORTC/MSGERC, but the corresponding reference is not 86.

*****RESPONSE:  Thank you for your comment and suggestions, as well as for noting these mistakes. These changes have been made as suggested.